# Gene Therapies in Dermatological Diseases: A Breakthrough in Treatment

**DOI:** 10.3390/ijms26146592

**Published:** 2025-07-09

**Authors:** Wiktoria Lisińska, Patryk Cegiełka, Zuzanna Zalewska, Natalia Bien, Dorota Sobolewska-Sztychny, Joanna Narbutt, Aleksandra Lesiak

**Affiliations:** 1Student Scientific Research Club of Experimental, Clinical and Procedural Dermatology, Medical University of Lodz, 90-419 Lodz, Poland; wiktoria.lisinska@stud.umed.lodz.pl (W.L.); patryk.cegielka@stud.umed.lodz.pl (P.C.); zuzanna.zalewska@stud.umed.lodz.pl (Z.Z.); 2Department of Dermatology, Pediatric Dermatology and Oncology, Medical University of Lodz, 90-419 Lodz, Poland; natalia.bien@umed.lodz.pl (N.B.); dorota.sobolewska-sztychny@umed.lodz.pl (D.S.-S.); joanna.narbutt@umed.lodz.pl (J.N.); 3Laboratory of Autoinflammatory, Genetic and Rare Skin Disorders at Department of Dermatology, Pediatric Dermatology and Dermatological Oncology, Medical University of Lodz, 90-419 Lodz, Poland

**Keywords:** gene therapies, dermatology, skin, psoriasis, RDEB, melanoma, wound healing

## Abstract

Gene therapy involves introducing genetic material into cells to treat or prevent disease and offers highly targeted and potentially curative approaches for both inherited and acquired conditions. The skin is an especially suitable organ for gene therapy due to its accessibility, ease of sampling, rapid cell turnover, and the possibility for localized treatment with minimal systemic exposure. Gene therapy is being actively explored across a range of dermatological conditions, including recessive dystrophic epidermolysis bullosa, ichthyosis, psoriasis, chronic wounds, and melanoma, with therapeutic strategies encompassing viral vectors, non-viral delivery systems, gene editing technologies, RNA-based treatments, and cell-based approaches. These diverse methods aim to correct genetic defects, modulate immune responses, promote tissue repair, or selectively target malignant cells. This review examines the advancements and potential of gene therapies in addressing complex skin diseases, providing hope for improved patient outcomes and long-term care.

## 1. Introduction

Gene therapy has its roots in the 1960s. The premise of these techniques is the introduction of external genetic material into cells lacking functional genes or containing defective genes. This involves techniques used to introduce, repair, replace, or silence genes in order to correct genetic defects or regulate abnormal gene expression. Key strategies in gene therapy include gene replacement, which compensates for missing or defective genes; gene silencing to reduce harmful gene expression; gene addition to introduce therapeutic genes; and gene editing technologies utilized for the precise modification of the genome. This genetic material can integrate into the cell’s DNA, and, following transcription, serves as a template for the production of the appropriate proteins [1]. To effectively deliver these therapies, various delivery systems are used, including viral vectors like lentiviruses, adenoviruses, and adeno-associated viruses (AAVs), which are efficient in targeting specific cells. Viruses intended to serve as carriers of genetic material are stripped of the genes responsible for virulence, thereby preventing viral replication within the host cell [2]. Retroviruses and adenoviruses are the viruses most used in gene therapy, although other viruses, such as herpesviruses and lentiviruses, can also be utilized. The viral vectors, such as the lentivirus, retrovirus, and HSV-1 used in RDEB and melanoma therapies, undergo rigorous quality control to confirm vector integrity, gene-insert accuracy, and the absence of replication-competent viruses [3,4,5]. Nonviral systems include polymer-based nanoparticles, lipid nanoparticles, and exosomes; these provide safer alternatives with lower immunogenicity. Stability studies typically assess gene expression retention over time, supported by longitudinal biopsy data and protein localization analyses in both animal models and human patients [6]. The skin consists of three layers—epidermis, dermis, and hypodermis—and is the largest and most accessible organ in the human body. This facilitates the efficient administration of topical therapies and monitoring of treatment outcomes [7]. The physiology of the skin is much more complex than had been previously thought. At present, we know that functions of the skin are regulated by local neuroendocrine and immune systems, by the means of multiple molecules, like cytokines, neurotransmitters, and neuropeptides. Dysfunctions in its functions, or the interactions between the local immune, hormonal, and endocrine systems and the microbiome, can lead to a wide variety of skin pathologies, such as inflammatory skin diseases like psoriasis, pigmentary disorders, or melanoma [8]. Interestingly, it has been noted that UVA radiation, and particularly UVB radiation, also influences the formation of skin neuro-immuno-endocrine reactions, which affect not only local skin homeostasis, but also the entire body. These responses involve, among others, the hypothalamic–pituitary–adrenal axis (HPA) and the regulation of the autonomic nervous system function [9].

Therefore, the skin is a very attractive organ for the development of novel therapeutics. In this review, we will focus on the updating of novel therapeutic approaches by the use of gene therapies in skin diseases.

## 2. Recessive Dystrophic Epidermolysis Bullosa (RDEB)

The dystrophic form of epidermolysis bullosa inherited in an autosomal recessive manner (recessive dystrophic epidermolysis bullosa, RDEB) is a rare subtype of epidermolysis bullosa (EB) [10]. The incidence of RDEB varies between populations; for example, it is 1 in 49,000 in Scotland and 1 in 420,168 in the United States [11]. This condition is caused by mutations in the *COL7A1* gene, which encodes type VII collagen, a component of anchoring fibrils (AFs). A defect in this gene disrupts the structure of the dermal–epidermal junction, leading to the formation of blisters on the skin. These blisters may be present at birth or appear later during development and can occur locally or be generalized. They form spontaneously or because of mechanical trauma. The recessively inherited form, which is the focus of this section, presents more severely than the dominant form (dominant dystrophic epidermolysis bullosa, DDEB) [4,12]. It has also been shown that individuals with RDEB have an up to 70-fold increased risk of developing squamous cell carcinoma, compared to the healthy population [13]. These findings, combined with the significantly reduced quality of life in RDEB patients and the lack of effective standard therapies, justify the pursuit of new treatment options [14].

Before the introduction of gene therapy, there was virtually no effective treatment for RDEB. Attempts at bone marrow transplantation have unfortunately proven unsatisfactory [4].

One breakthrough in RDEB therapy has been the development of topical gene therapy. Beremagene geperpavec (B-VEC) is a non-replicating viral vector used to deliver two copies of the *COL7A1* gene directly into skin cells (Figure 1A). The delivery of viruses has been associated with high biological purity and without the genes responsible for virulence, which had previously been removed from the viruses. The technology utilizes herpes simplex virus type 1 (HSV-1), which is capable of carrying large genes such as the one for type VII collagen (C7). This method was initially tested in vitro on skin cells; this was followed by intradermal injection of the virus in mice with RDEB-like features. The studies showed that B-VEC successfully delivered the gene, and the synthesized protein localized correctly in the skin. These findings were observed both in mouse models and human xenografts. The efficacy of the topical application was later assessed in a randomized, placebo-controlled phase 1/2 clinical trial in humans. The product demonstrated a profile of very high safety, without any serious or significant adverse effects. In a 3-month evaluation, almost all patients showed wound healing, while the placebo group exhibited inconsistent wound healing and the formation of new blisters [4,15]. In phase 3 trials involving 31 patients (both children and adults), two morphologically similar wounds were selected per patient. B-VEC was applied to one wound and placebo to the other. Again, high efficacy was demonstrated: 67% of wounds treated with gene therapy healed, compared to 22% in the placebo group. None of the observed serious adverse events were related to the medicinal product [16]. In 2023, the drug received FDA approval [17]. On April 2025, the drug (brand name—Vyjuvek) was approved by the European Medicines Agency (EMA) [18].

Efforts have also been made to introduce genetically modified cells into RDEB patients through the use of viral vectors. One example is lentivirus-modified fibroblasts delivered via intradermal injections (Figure 1B). In one study, patient-derived fibroblasts were transduced with a lentiviral vector carrying the *COL7A1* gene, which was achieved by transduction of the entire length of the COL7A1 cDNA codon under the control of the human phosphoglycerate kinase (PGK) promoter and then reintroduced into the patient via intradermal injection. However, before studies were conducted in a human model, preclinical studies in mouse models showed the presence of C7 at the dermal–epidermal junction. Each patient received three injections on the first day of therapy [14]. Adverse effects included erythema, pain, and bruising at the injection site. Some moderate-to-severe complications were reported during treatment that were not necessarily related to the therapy, and no life-threatening events occurred. Collagen VII expression was evaluated at 3 weeks, 3 months, and 12 months post-injection. All four participants underwent 12 skin biopsies to assess protein expression. A significant increase in protein levels was observed in half of the samples, though fully developed anchoring fibrils were not visualized [5].

Another approach tested the effectiveness of transplanting genetically modified keratinocytes using a retroviral vector carrying the *COL7A1* gene (Figure 1C). This study, conducted at Stanford University, included four adult patients with genetically confirmed RDEB, each having wound areas of at least 100 cm^2^ suitable for grafting. Keratinocytes were initially biopsied and cultured. They were then infected with a virus carrying the gene encoding the collagen molecule, and the entire process was controlled by the Moloney leukemia virus long terminal repeat promotor. Several weeks later, grafts were applied to six wounds per patient, including one induced by rubbing. Gene expression was assessed at 3, 6, and 12 months post-grafting, along with wound healing. No serious adverse events were observed. The most common side effect was an itching near the treated area, and no systemic autoimmune reactions were detected. Collagen expression was found in 90% of skin biopsies at 3 months, 66% at 6 months, and 40% at 12 months, with correct localization at the dermal–epidermal junction. Fully formed anchoring fibrils were identified in 71% of biopsies at 3 months and 33% at 6 months [19]. Although these results are promising, one negative aspect was the decline in collagen expression after one year. In a phase 1/2a study involving seven patients (including the four described above), 90% of wounds healed after 6 months, compared to 0% in control wounds. After 3 years, 80% wound healing was observed. After 12 months, 43% of patients showed developed AFs; after 2 years, two out of three evaluated patients showed AFs. The most common adverse effects were infections, itching, and pain around the wound [20]. Long-term studies showed ≥50% wound healing in 70% of patients five years post-transplant, along with significant pain reduction (by about 20 percentage points compared to pre-treatment). No serious adverse events directly related to the therapy were observed, though mild-to-moderate side effects did occur, including the occurrence of antibodies against transplanted keratinocytes and localized itching. Two patients died due to disease progression [21]. A drug using the above mechanism, named Zevaskyn, was approved by the FDA on 29 April 2025 [22].

As these studies show, various gene therapy methods are being explored with varying degrees of success. Currently, beremagene geperpavec appears to be the most thoroughly studied and effective treatment option.

## 3. Melanoma

Melanoma is one of the most dangerous skin cancers. Although it accounts for only 1% of diagnosed skin cancers, it is unfortunately often associated with an unfavorable prognosis. The basic method of treating melanoma is its removal with an appropriate margin of healthy tissue, if necessary, including the affected lymph nodes. Treatment also includes radiotherapy, chemotherapy and various methods of immunotherapy. Recently, gene therapies have joined the range of therapeutic options [23].

### 3.1. Small Interfering RNA (siRNA)-Based Treatment

Small interfering RNA (siRNA) is a type of double-stranded RNA consisting of 21–23 nucleotides. The siRNA induces gene silencing by inhibiting gene expression. The antisense strand of siRNA leads to the degradation of a complementary mRNA molecule through the action of a ribonuclease-active protein complex called RISC. As long as RISC remains active, gene expression is suppressed [24]. One of the greatest challenges in siRNA-based therapies is finding the most effective method for delivering therapeutic nucleic acids to the cell nucleus. The siRNAs on their own are highly unstable and very sensitive to endogenous nucleases. Various carriers are used to deliver siRNA to tumor cells, including lipids, polymers, liposomes, and exosomes [25]. Currently, there are no approved gene therapies using siRNA. Therefore, in the section below, we present what we believe to be the most promising approaches for treating melanoma.

In a study by Xueyan Zhang, Anqi Cai et al., the therapeutic target was the WEE1 gene, the protein product of which is a kinase playing a key role in the G2/M checkpoint by phosphorylating CDC2 protein. Overexpression of WEE1 has been observed in various types of cancer, including melanoma. The authors developed a unique delivery system—RRCPP—composed of LMW PEI as the base, a cell-penetrating peptide (R8), cholesterol, PEG, and cyclic RGD (which binds to integrin αvβ3), focusing on the treatment of melanoma lung metastases. The study used the following cell lines: B16F10 (mouse melanoma), C26 (mouse colon cancer), 293T (human embryonic kidney cells), and 4T1 (mouse breast cancer). C57 female mice were utilized. The results showed that RRCPP nanoparticles stably transport siRNA and are safe and effective. Additionally, the RRCPP/siWee1 complex significantly inhibited tumor growth in melanoma models, indicating its potential as a gene therapy in oncology. Compared to the control group (NS), the RRCPP/siWee1 complex significantly delayed B16 tumor growth, achieving an inhibition rate of nearly 85.2% based on mean tumor volume (*p* < 0.001), whereas RRCPP/siNC treatment led to only slight tumor-growth suppression. Intravenous administration of the drug in the metastatic lung tumor model showed significant reductions in tumor mass and the number of metastatic tumors (compared to the control group), without an increase in side effects compared to the control group [26].

Another interesting direction is the combination of widely used and established therapeutic methods with siRNA. In a study by Changrong Wang, Xiaoguang Shi et al., a polymer–lipid delivery system was used to transport both siRNA and the chemotherapeutic drug doxorubicin. The siRNA used inhibited PD-1 protein expression, and, in combination with doxorubicin, increased the T lymphocyte count and immune activity against melanoma cells—paving a new path for immunotherapy. This combination proved effective in treating melanoma, both prophylactically and in metastases. This regimen was more effective than monotherapies and ensured 100% animal survival during treatment, with good in vivo safety [27].

Similar conclusions were reached by Chenyang Li, Xiuping Han et al., who placed both siRNA and imatinib in liposomal nanoparticles. Their combination synergistically silenced PD-1 expression, restoring the cytotoxic activity of T cells against cancer cells and inhibiting the pro-tumorigenic mTOR pathway. This led to tumor-growth delay and increased INF-γ levels in the lymph nodes and the spleen. In the in vitro study results, decreased PD-1 and p-S6K levels were observed in B16F10 cells, along with an increase in tumor cell apoptosis to approximately 60% (compared to about 35% when siRNA and imatinib were used separately). In the in vivo studies, which used a mouse model, a slowdown in tumor growth was observed compared to the control group, along with an increase in IFN-γ levels in the lymph nodes and the spleen, confirming the immune response. Additionally, as in the in vivo studies, tumor growth was more effectively inhibited than when siRNA and imatinib were used separately [28].

Apart from the question of delivery methods utilized for siRNA-based drugs within the body, researchers are also exploring how to administer them. These substances are mostly administered intravenously or intramuscularly. In the study by Ramos-Gonzalez, Martin R et al., in addition to in vivo testing with the B16F10 melanoma cell line, in vitro studies were conducted using siRNA, silencing the WT1 protein in melanoma lung metastases; this was delivered via an innovative respiratory route. After two weeks of treatment, WT1 silencing showed strong anti-tumor effects, leading to a reduction in tumor mass and significantly delaying animal death [29].

In study by Ruan R., Chen M. et al., a topical peptide-based carrier called SPACE-EGF was used, containing siRNA targeting the c-Myc and GAPDH genes. In vivo studies showed that siRNA could be locally delivered through the skin and effectively transported to target tumor cells, leading to gene silencing and apoptosis induction [30].

All of the currently available data on siRNA therapy in melanoma come from preclinical studies. At present, there are no ongoing clinical trials using siRNA for the treatment of melanoma.

### 3.2. CAR-T

CAR-T therapy is a type of cell and gene therapy. CAR-T cells (chimeric antigen receptor T cells) are genetically modified T lymphocytes designed to locate and destroy cancer cells. Unlike regular T cells, CAR-T cells can recognize receptors on the surface of tumor cells without the need for MHC molecules or co-receptors. This is made possible by a chimeric receptor composed of an antigen-binding domain, a signaling domain, and a transmembrane domain that links the two. The antigen-binding domain is derived from immunoglobulin fragments, enabling the recognition of tumor cell antigens independently of MHC molecules. The signaling domain contains a portion of the CD3ζ glycoprotein from the T-cell receptor and optionally includes a co-receptor signaling domain, eliminating the need for co-receptors to activate CAR-T cells. So far, this therapy is primarily used to treat hematologic cancers, but research is ongoing into its use for solid tumors and autoimmune diseases [31,32].

Applying CAR-T therapy to solid tumors, including melanoma, remains a challenge. Identifying tumor-specific antigens is crucial, along with developing methods to ensure CAR-T cells effectively penetrate the tumor microenvironment and overcome the immunosuppressive tumor milieu (TME) [33]. Potential therapeutic targets identified in melanoma include VEGFR-2, CD-16, CD-70, HER-2, and B7-H3. One major obstacle is the heterogeneity of antigen expression in melanoma cells. Despite promising preclinical results, clinical trials targeting metastatic melanoma have not met expectations [34]. In one study involving a chimeric receptor against VEGFR-2 administered with IL-2, the clinical phase was discontinued due to a lack of therapeutic response and adverse effects in most patients [35].

In February 2024, a study was published describing the use of CAR-T therapy to target cancer cells with high TRP1 protein expression (a tyrosinase-related peptide) in a population of patients with rare and checkpoint inhibitor-resistant melanoma subtypes. Although TRP1 is an intracellular melanosome protein, a small fraction reaches the cell surface. Its elevated expression has been observed in acral melanoma, mucosal melanoma (60%), and choroidal melanoma (90%).

This therapy showed a high affinity for tumor cells, without harming normal cells expressing TRP1, thus minimizing the side effects. The study was conducted both in vitro and in vivo. The results showed anti-tumor activity, and in immunocompetent mouse models, no serious systemic toxicity or off-tumor effects were observed [36]. CAR-T therapy represents a promising direction for new melanoma treatments, but further research is needed.

### 3.3. Oncolytic Viruses

Oncolytic virus therapy combines features of immunotherapy and gene therapy. Oncolytic viruses replicate within cancer cells, causing their lysis and triggering a systemic immune response [37,38]. One FDA-approved oncolytic viral therapy for melanoma is Talimogene laherparepvec (T-VEC), approved in 2015. T-VEC is a genetically modified live virus derived from herpes simplex virus type 1 (HSV-1) [39]. Genetic engineering removed two viral genes, ICP34.5 and ICP47, reducing the virus’s ability to infect healthy cells and increasing its tumor selectivity. Additionally, a human GM-CSF transgene was inserted to stimulate immune responses both locally and systemically [40]. T-VEC is reserved for patients with advanced, unresectable melanoma [41]. It has shown efficacy in stage III and stage IV melanoma, improving durable remission rates and disease-free survival by 50% over 60 months. Nearly all patients who achieved remission remained disease-free during the five-year follow-up. Research is ongoing on another HSV–1-based oncolytic virus—HF10—used in combination therapies [37].

## 4. Psoriasis

Psoriasis is a chronic, inflammatory skin disease characterized by localized or extensive skin thickening with hyperproliferation, scaling and erythema [42], abnormal differentiation, and impaired apoptosis of keratinocytes, alongside excessive immune cell infiltration and pathological angiogenesis. It is often associated with comorbidities like psoriatic arthritis, cardiovascular disease, and mental health issues, significantly impacting patients’ quality of life [43].

There is no clear consensus on the treatment of psoriasis. According to the traditional approach, the treatment options can be divided based on the form of the disease. The standard treatments for mild psoriasis are topical corticosteroids, vitamin D analogues, and calcineurin inhibitors. Moderate-to-severe forms are treated with, for example, methotrexate, acitretin, phototherapy, or biological drugs [44].

The pathogenesis of psoriasis is characterized by a complex interplay of chronic inflammation, aberrant angiogenesis, and dysregulated keratinocyte proliferation and differentiation. As a result, therapeutic interventions are aimed at modulating these interconnected pathological processes [45]. Since aberrant keratinocyte proliferation and differentiation serve as key pathogenic features in psoriasis, therapeutic options have shown efficacy in both clinical and animal models. Antisense oligonucleotides and siRNAs targeting keratin 17 (K17), which is a type I intermediate filament providing mechanical support for keratinocytes in order to maintain the functional integrity of the epidermis, have proven effective in reducing keratinocyte proliferation and inducing apoptosis in preclinical studies [46]. Moreover, the modulating of genes such as *FGFR2*, *NFAT2*, *TRAF3IP2*, *AKR1B10*, and *POMP* has been shown to effectively suppress keratinocyte proliferation. For example, *FGFR2* inhibits keratinocyte proliferation and decreases epidermal thickness, and *TRAF3IP2* knockdown diminishes the proliferation of keratinocytes and endothelial cells by promoting apoptotic signaling and blocking the G2/M cell cycle phase [47]. Moreover, genomic targeting approaches toward interrelated gene networks such as *WT1* [48], *WTAP*, *EGR1*, and *PLK2* suggests that upstream targets may offer broader therapeutic benefits due to collaborative participation in cell growth and differentiation [49]. Aside from keratinocyte inhibition, siRNA-mediated silencing of the *GRHL2*, *SGPL1*, and *CTSB* genes restores normal keratinocyte differentiation [50].

Meanwhile, inflammation signaling could be corrected by promoting miR-125b or inhibiting miR-31 and miR-210 (Table 1). Simultaneous silencing of multiple psoriasis-related genes such as *DEFB4*, *TSLP*, and *KRT17* also reduces the level of inflammatory markers [49]. Tumor necrosis factor-alpha (TNFα) is a key driver of inflammatory responses in psoriasis, although systemic biologic therapies effectively targeting TNFα are expensive and have many adverse effects. As a result, siRNAs targeting TNFα, topically applied by chitosan-based vectors, are promising topical siRNA-based therapies. Chitosan is chemically modified by grafting diisopropylethylamine (DIPEA) and polyethylene glycol (PEG) onto polysaccharide to enhance siRNA stability, facilitate skin penetration, and ensure safe and effective delivery under physiological conditions, ensuring protection against RNase degradation and oxidative stress. In vitro studies confirmed not only that the polyplexes demonstrated low cytotoxicity in both keratinocytes and fibroblasts and achieved TNFα knockdown efficiencies of up to 65%, but also that the chitosan vectors significantly enhanced siRNA delivery compared to unformulated siRNA. Moreover, in vivo experiments revealed that topically applied DIPEA-chitosan/siRNA polyplexes preferentially accumulated in hair follicles, facilitating deeper skin penetration. Treatment led to an approximately 50% reduction in local TNFα expression, decreased infiltration of immune cells, and partial restoration of normal epidermal architecture, highlighting the therapeutic promise of this approach [51,52]. Importantly, siRNA-mediated knockdown of *CK2* not only corrects differentiation defects but also reduces inflammatory cytokine production driven by IL-17A. A recent study by Mandal et al. demonstrated that ionic liquids can effectively deliver NFKBIZ siRNA into the skin, successfully suppressing abnormal gene expression and reducing key psoriasis-related inflammatory markers like TNF-α and IL-17A [53]. Similarly, Lee et al. utilized the ablative laser-assisted delivery of nanocarriers to transport IL-6 siRNA, which effectively alleviated the psoriasis [54].

Signal transducer and activator of transcription 3 (STAT3) is a transcription factor essential for Th17 cell activation and inflammatory signaling receptors such as IL-6, IL-21, and IL-23. Moreover, STAT3 is constitutively activated and plays an important role in epidermal keratinocytes of human psoriatic skin lesions [56]. However, targeting STAT3 with small molecules has been proven to encounter difficulties due to its non-catalytic nature. As a result, siRNA targeting STAT3, delivered via lipid nanoparticles (LNPs), offers a promising alternative treatment strategy. LNPs are efficient and clinically validated delivery systems, but their high immunogenicity, attributed to ionizable cationic lipids components, can cause excessive inflammation mediated by release of pro-inflammatory cytokines like TNF-α, IL-6, and IL-1β [57]. To address this, researchers developed a modified LNP containing a fifth anti-inflammatory lipid, enhancing delivery while reducing immune responses. Researchers have been advancing lipid nanoparticle (LNP) formulations to enhance the safety and efficacy of nucleic acid therapeutics, focusing on optimizing lipid ratios, developing novel ionizable lipids, and improving tissue-specific targeting and intracellular delivery. A recent study introduced an innovative approach by incorporating budesonide, an anti-inflammatory agent, into a four-lipid LNP system to mitigate inflammation while preserving efficient siRNA delivery [58]. The modified LNPs, termed C8B2-si-STAT3, demonstrated superior anti-inflammatory effects compared to conventional LNPs, effectively reducing inflammatory mediators and alleviating psoriasis symptoms in a murine model [52]. This formulation exhibited favorable biocompatibility, with no observed toxicity in major organs, suggesting its potential for treating psoriasis and other inflammatory disorders, including rheumatoid arthritis and inflammatory bowel disease. The refined C8B2 system offers a promising platform for long-term RNA interference (RNAi) therapies, minimizing immune-related side effects and supporting scalable production, thereby positioning it for future applications in combination therapies. Efforts are ongoing to further enhance its stability and enable consistent, large-scale manufacturing [59].

On the other hand, abnormal angiogenesis is another critical feature of psoriasis. Angiogenesis is a crucial process in the development of skin lesions; hence, therapies modulating angiogenesis are under investigation. The siRNA-based targeting of *AQP1*, *HIF-1α*, *TRAF3IP2*, and *KRT16* reduces VEGF expression and disrupts pro-angiogenic signaling pathways such as ERK and PI3K/MTOR, reducing new blood vessel formation [60,61]. Collectively, these findings highlight the multi-faceted role of siRNA in regulating keratinocyte behavior and angiogenesis, providing a versatile platform that enables several pathological aspects of psoriasis to be addressed.

While various treatments exist to manage psoriasis, none provide a definitive cure. However, advances in biologics and gene therapy are reshaping treatment strategies, offering the potential for longer-lasting disease control and symptom relief [62]. While these results are largely based on preclinical studies, they underscore the significant potential of siRNA-based therapies, pending the further development of efficient delivery systems and validation in clinical trials [44].

## 5. Wound Healing

Wound healing is a multifaceted biological process involving orchestrated phases of hemostasis, inflammation, proliferation, and remodeling. However, in chronic wounds such as diabetic foot ulcers (DFUs), venous leg ulcers (VLUs), and pressure ulcers (PUs), this process is disrupted, leading to persistent inflammation, poor angiogenesis, and impaired tissue regeneration [63]. So far, in the treatment of chronic wounds, negative pressure therapy, biosurgery, and ultrasound therapy have been used [64]. Gene therapy has emerged as a promising strategy used to address impaired growth factor (GF) signaling in chronic wounds, offering an alternative to exogenous GF delivery. While topical GF formulations such as Regranex (PDGF), Fiblast (FGF), and Heberprot-P (EGF) have shown potential—with some achieving FDA approval or advancing to clinical trials—their clinical success has been limited by GF instability in the wound environment and the high doses needed for therapeutic effect, factors that raise safety concerns. To overcome these challenges, gene delivery systems that promote sustained endogenous GF expression have gained interest [65]. Viral vectors have shown high efficiency and progressed in early clinical trials but face hurdles like immunogenicity and the risk of insertional mutagenesis, impeding broader clinical use. Viral vectors offer high gene transfer efficiency and the ability to target nondividing cells, making them attractive for delivering growth factors (GFs) in chronic wound healing. AAVs are the most studied, showing promise with PDGF and VEGF delivery, but inconsistent results, and safety concerns have hindered clinical approval. In contrast, nonviral approaches—primarily lipid- or polymer-based—offer improved safety and design flexibility but struggle with low transfection efficiency. Additionally, most existing therapies target only a single GF, which may be insufficient for effective wound healing, highlighting the need for multi-GF gene therapy solutions. Current research focuses increasingly on nonviral gene delivery platforms capable of regulated, multi-GF expression aiming to enhance therapeutic outcomes in chronic wound treatment [66].

Antisense oligonucleotides (ASOs) and messenger RNA (mRNA) represent a transformative class of modalities for the treatment of skin disorders, offering targeted and adaptable approaches to the modulation of gene expression, as seen in Figure 2. RNAi harnesses the endogenous gene-silencing machinery via siRNAs or miRNAs that guide the RNA-induced silencing complex (RISC) to degrade or suppress the translation of specific mRNAs, effectively downregulating pathogenic proteins. This has proven especially valuable in fibrotic skin diseases, in which siRNAs targeting CTGF (Connective Tissue Growth Factor)—such as BMT101, LEMS401, and RXI-109—are undergoing clinical evaluation for their ability to reduce hypertrophic scarring and keloid formation. Currently, among siRNA-based studies targeting the skin, four therapeutics have advanced to clinical trials: STP705 (cotsiranib) delivered via nanoparticles, along with the naked siRNA formulations BMT101, OLX10010, and RXI-109. The goals of these siRNA drugs are the same: the treatment of hypertrophic scars caused by the excessive production of collagen from myofibroblasts during wound healing. All these siRNAs are delivered to the skin by intradermal injection. BMT101 (cp-asiRNA), OLX10010, and RXI-109 target connective tissue growth factor (CTGF), which is involved in the formation of hypertrophic scars and keloids. STP705 targets TGFβ1 and Cyclooxygenase-2 (COX-2), which modulate signaling pathways related to hypertrophic scars [59]. In contrast, ASOs are short, single-stranded nucleic acids that bind complementary RNA sequences to either trigger degradation through RNase or sterically hinder transcript processing. With chemically engineered generations improving their stability and affinity, ASOs like MRG-110, which targets miR-92a to promote angiogenesis via ITGA5 upregulation, have entered clinical trials for uses involving the enhancement of wound healing [67]. Each of these RNA-based platforms offers distinct advantages: RNAi and ASOs provide high target specificity and gene-silencing precision, while mRNA allows transient protein expression and rapid vaccine design. A study conducted by Forouzan K. has also demonstrated the therapeutic potential of miR-192 in promoting chronic wound healing. As mentioned above, miR-192 is a microRNA known to regulate genes involved in collagen production and the TGF-β/Smad signaling pathway, which is crucial in tissue regeneration. Overexpression of miR-192 in fibroblast cells led to a significant increase in COL1A2 expression, enhancing collagen synthesis and supporting fibroblast proliferation and migration [68]. When these engineered cells were combined with platelet-rich plasma (PRP), the wound healing response in diabetic rats was markedly improved [69]. The combination treatment accelerated wound closure, increased granulation and epithelial tissue formation, and promoted angiogenesis. These results suggest that miR-192 plays a central role in modulating the wound healing environment and could serve as a key target in regenerative therapies for chronic wounds [70].

Together, these approaches form a complementary arsenal poised to address both monogenic and multifactorial skin disorders. However, challenges remain, including issues of delivery, off-target effects, and tissue penetration. Continued innovations in RNA chemistry and delivery technologies will be critical to fully realize their therapeutic potential in dermatology [68].

## 6. Ichthyosis

Inherited ichthyotic conditions are a group of rare genetic disorders primarily affecting the skin, and typically presenting with generalized dryness, scaling, hyperkeratosis (thickening of the skin), and varying degrees of erythroderma (skin redness). These disorders are broadly classified into syndromic forms, which involve multiple organ systems, and non-syndromic forms, which are limited to the skin. One major group of non-syndromic ichthyoses is autosomal recessive congenital ichthyosis (ARCI), which is genetically and phenotypically diverse and includes subtypes such as lamellar ichthyosis (LI), harlequin ichthyosis (HI), and congenital ichthyosiform erythroderma (CIE) [71]. LI is linked to TGM1 mutations and typically manifests at birth with a collodion membrane that evolves into thick, brown, plate-like scales; mild erythroderma and impaired sweating due to sweat duct obstruction are common. So far, biologic drugs have been used in the treatment of ichthyosis spectrum disorders, including secukinumab (antibody against IL-17A), ixekizumab (against the IL-17A receptor), ustekinumab (against IL-12 and IL-23) and dupilumab (blocking IL-4 and IL-13). However, the data on their efficacy come mainly from clinical case reports and are unsatisfactory in some cases [72]. As LI is genetically monogenic and the skin is an accessible target tissue, LI has been the focus of several preclinical and clinical gene therapy efforts. Researchers have tried to restore *TGM1* expression and skin barrier function in patient-derived keratinocytes by retroviral vectors, for example, by using TGM1-carrying self-inactivating γ-retroviral (SINγ-RV) under the control of its own promoter, and thereby obtaining appropriate TG1 expression in keratinocytes. More recently, Freedman et al. developed KB105, a modified herpes simplex virus type 1 (HSV-1) vector encoding human *TGM1*, which successfully restored *TGM1* expression in vivo without causing adverse effects such as fibrosis or inflammation; this therapy is currently undergoing clinical evaluation [73]. Moreover, among the viral delivery systems, herpes simplex virus type 1 (HSV-1) vectors have demonstrated substantial promise, owing to their natural tropism for epidermal keratinocytes and their non-integrating nature, which minimizes the risk of insertional mutagenesis [74]. Another promising approach, used adenine base editors (ABEs) and single-guide RNAs (sgRNAs) to correct *TGM1* mutations in human zygotes, providing foundational data for potential in utero therapies. In contrast, HI is the most severe form of ARCI, clinically presenting with thick, armor-like scales and severe facial malformations, including ectropion (outward turning eyelids), eclabium (outward turning lips), and underdeveloped ears. HI is caused by loss-of-function mutations in *ABCA12* (ATP binding cassette subfamily A member 12) that impair the lipid transport in lamellar granules of keratinocytes during skin development. In 2005, Akiyama et al. not only elucidated HI’s genetic basis but also demonstrated partial rescue of lipid secretion via gene transfer using full-length ABCA12 cDNA [70]. Complementary strategies such as autologous grafting of genetically corrected keratinocytes have also shown potential, particularly in analogous genodermatoses such as Netherton syndrome. These approaches offer the advantage of a localized, patient specific therapy that circumvents systemic immune complications. Furthermore, gene editing technologies—including transcription activator-like effector nucleases (TALENs) and allele-specific small interfering RNAs (siRNAs)—have enabled the targeted correction or silencing of dominant-negative mutations in epidermolytic ichthyosis and keratitis–ichthyosis–deafness (KID) syndrome, suggesting that similar methods could be extended to recessive forms of ichthyosis where gain-of-function alleles are not involved [75]. Given the devastating clinical presentation of HI and the underlying monogenic defects, extensive translational research efforts continue to focus on developing targeted gene therapies for both LI and HI. Despite these advancements, several key challenges remain [71]. The structural and functional complexity of the skin, combined with the high turnover of keratinocytes and the need to target epidermal stem cell populations for sustained correction, complicates durable therapeutic outcomes. Furthermore, the rarity and genetic heterogeneity of ichthyosis subtypes create barriers to the conducting of large-scale clinical trials. Immune responses to vectors, particularly in the context of repeated administration, must also be carefully managed.

## 7. Conclusions

The skin’s accessibility and regenerative capacity make it an exceptionally attractive target for gene therapy. Emerging therapeutic strategies may involve the transplantation of cells transduced with viral vectors encoding genes absent in the host, as well as the use of encapsulated nucleic acids—commonly employed in mRNA-based therapies and polymer-based delivery systems (e.g., polyethyleneimine) for plasmid DNA administration. Gene therapy is being used in the treatment of a rare disease called recessive dystrophic epidermolysis. At least a few of the methods presented in this text have the potential to become a real help for patients suffering from this rare disease. As emphasized, the most promising method is the one using B-VEC, one which, which is also important, has shown high efficacy and a high level of safety, resulting in registration by the FDA. As to melanoma, gene therapy has broadened its reach from the targeting of oncogenic drivers to stimulating immune responses. Strategies like siRNA-mediated gene silencing, CAR-T cells, and oncolytic viruses (e.g., T-VEC) are under investigation or in clinical use, employed to overcome tumor immune evasion, however, barriers such as tumor heterogeneity and immune resistance still require further innovation. Psoriasis, a multifactorial inflammatory condition, has prompted a growing interest in RNA-based therapies, particularly siRNA and miRNA approaches targeting STAT3, keratin 17, and TNF-α pathways, which aim to normalize keratinocyte behavior, suppress inflammation, and modulate angiogenesis. Innovative delivery methods such as chitosan-based nanocarriers, lipid nanoparticles, and laser-assisted systems have enhanced the feasibility of topical gene therapies in psoriatic skin. In the field of wound healing, especially in chronic wounds such as diabetic foot ulcers and pressure sores, gene therapy includes the use of growth factor-encoding vectors, mRNA, antisense oligonucleotides (ASOs), and miRNA-based therapies. Additionally, LNPs have been utilized as vehicles for gene and drug delivery to improve burn wound healing, demonstrating the manner in which innovations derived from genodermatoses research are extending to broader therapeutic applications. And finally, inherited ichthyoses, particularly lamellar and harlequin ichthyosis, which are rare monogenic skin disorders, have become promising targets for gene therapy, with advances in viral vectors, gene editing, and cell-based approaches showing encouraging preclinical and early clinical results despite remaining challenges in delivery, durability, and trial scalability (Table 2).

While most of these approaches are in early development or clinical trial phases, the growing evidence supports the potential of gene therapy to provide durable and targeted treatments for skin diseases. Continued innovation in vector design, delivery systems, and clinical translation will be key to realizing the full therapeutic impacts of these strategies.

This review has limitations. There were no specific selection criteria or research methodologies used to identify the sources and references for this review. In addition, due to the extent of the topic, the focus was on several major gene therapies and their potential roles in the treatment of skin disease. The review also has strengths. We have presented overviews for the most recent update on gene therapies in RDEB, melanoma, psoriasis, and wound healing.

## Figures and Tables

**Figure 1 ijms-26-06592-f001:**
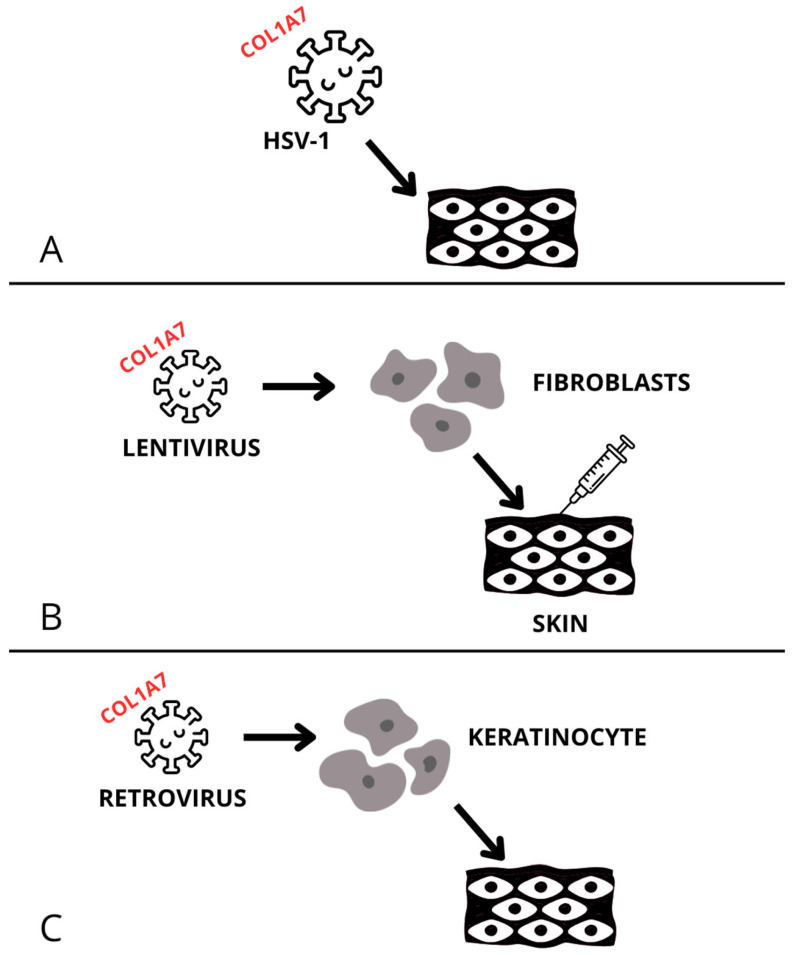
(**A**) This figure presents a model of the use of B-VEC. (**B**) This figure presents an overview of the therapy, consisting of collecting fibroblasts from patients, infecting them with a lentivirus carrying the COL1A7 gene, and then subcutaneously injecting the infected cells. (**C**) This figure presents a method of transplanting retrovirus-infected keratinocytes.

**Figure 2 ijms-26-06592-f002:**
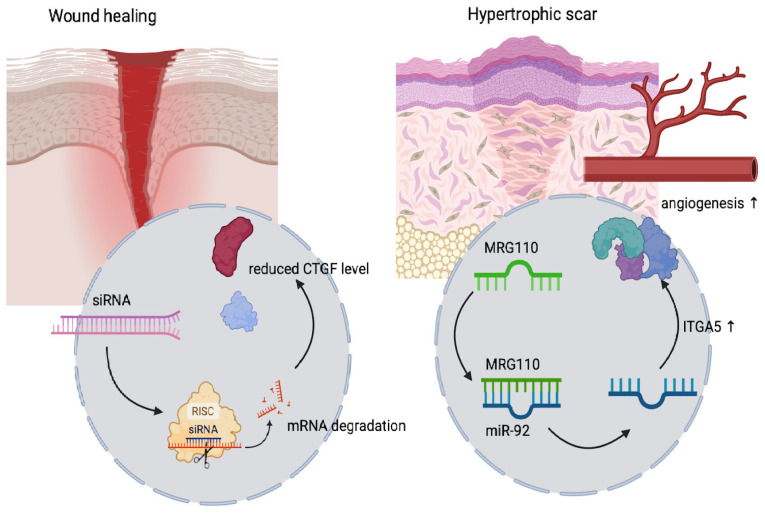
iRNAs targeting CTGF mRNA aim to reduce CTGF protein levels, and as a result, fibrosis and scarring in hypertrophic scars. MRG-110 targets *miR-92a*, a negative regulator of *integrin alpha 5* (ITGA5). Inhibiting *miR-92a* leads to increased ITGA5 expression, enhancing angiogenesis and promoting wound healing. CTGF—Connective Tissue Growth Factor; MRG-110—Antagomir Targeting miR-92a, ITGA5—integrin alpha 5, miR-92a—mi-RNA 92a.

**Table 1 ijms-26-06592-t001:** Roles of miRNAs and siRNAs as potential targets of novel treatments for psoriasis [55].

RNA Name	Target Gene/Pathway	Role in Psoriasis
miR-155	SOCS1	Promotes inflammation via Th17 pathway
miR-340	STAT3	Inhibits keratinocyte hyperproliferation
miR-21	Multiple (e.g., TGF-β, immune regulators)	Enhances inflammation and epidermal thickening
miR-31	PP6	Promotes keratinocyte proliferation; drives inflammation via NF-κB and IL-1β/Th17 axis
IL-6 siRNA	IL-6	Reduces inflammatory cytokine production
K17 siRNA	Keratin 17	Lowers keratinocyte activation and inflammation
Pcsk9 siRNA	PCSK9	Regulates lipid metabolism and inflammatory signaling
TNF-α siRNA	TNF-α	Decreases TNF-α expression; reduces skin inflammation

miR-155—miRNA-155; SOCS1—suppressor of cytokine signaling 1; Th17—T helper 17 cells; miR-340—miRNA-340; STAT3—signal transducer and activator of transcription 3; miR-21—miRNA-21; TGF-β—transforming growth factor beta; miR-31—miRNA-31; PP6—protein phosphatase 6; NF-κB—nuclear factor kappa-light-chain-enhancer of activated B cells; IL-1β—interleukin 1 beta; IL-6 siRNA—interleukin-6 small interfering RNA; IL-6—interleukin-6; K17 siRNA—keratin 17 small interfering RNA; Pcsk9 siRNA—proprotein convertase subtilisin/kexin type 9 small interfering RNA; PCSK9—proprotein convertase subtilisin/kexin type 9; TNF-α siRNA—tumor necrosis factor alpha small interfering RNA; TNF-α—tumor necrosis factor alpha.

**Table 2 ijms-26-06592-t002:** Summary of the diseases, therapeutic methods, and publications described in this review.

**Disease**	**Summary of Gene Therapy Treatments**	**References**
RDEB (Recessive Dystrophic Epidermolysis Bullosa)	Topical gene therapy (B-VEC)	[4,15,16,17,18]
	Transplanting genetically modified keratinocytes using a retroviral vector	[19,20,21,22]
	Intradermal injections of lentivirus-modified fibroblasts	[5,14]
Melanoma	siRNA-based treatment (WEE1 and WT1 genes as a therapeutic target, siRNA combined with immunotherapy and siRNA delivered via the peptide-based carrier SPACE-EGF)	[24,25,26,27,28,29]
	CAR-T therapy	[31,32,33,34,35,36]
	Oncolytic virus therapy (T-VEC and H10)	[37,38,39,40,41]
Psoriasis	Antisense oligonucleotides and siRNA targeting keratin 17 (K17)	[46]
	siRNA targeting inflammatory factors (e.g., TNFα, IL-6, and PCSK9)	[42,43,44,45]
	siRNA inhibiting keratinocyte proliferation and improving the regulation of cell growth and differentiation (e.g., FGFR2, NFAT2, WT1, and WTAP)	[48,49]
	siRNA reducing VEGF expression and inhibiting angiogenesis (e.g., AQP1, HIF-1α, TRAF3IP2, and KRT16)	[59]
Wound Healing	siRNA based treatment (TGFβ1, COX-2 and CTGF as a therapeutic targets)	[63]
	Therapies using viral and non-viral vectors as a delivery system	[64,69]
	Antisense oligonucleotide-based therapy (miR-92a as a target)	[70]
	mRNA- and miRNA-based therapies	[68]
Ichthyosis	Restoring TGM1 expression via viral vectors	[73]
	Adenine-base editors and single-guide RNAs used to correct TGM1 mutation	[73]
	Gene transfer using full-length ABCA12 cDNA	[72,74]
	Autologous grafting of genetically corrected keratinocytes	[72]
	Gene editing using transcription activator-like effector nucleases (TALENs) and allele-specific small interfering RNAs (siRNAs)	[75]

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
