# Peer review of "Gene Therapies in Dermatological Diseases: A Breakthrough in Treatment"

_ijms, 2025, doi:10.3390/ijms26146592_

Round 1

Reviewer 1 Report

Comments and Suggestions for Authors

Dear authors,

The manuscript entitled “Gene therapies in dermatological diseases: an advance in treatment” is a review focused on the new therapeutic approaches with gene therapies in skin diseases, presenting the clinical and academic scenario.

The manuscript is relevant and modern, and when reading it, some questions arose. I would like to obtain information from the authors regarding the following:

- How is the stability and purity study of these therapeutic genes conducted?

- What are the main attributes analyzed in the quality control test of these therapeutic genes?

Reviewer 2 Report

Comments and Suggestions for Authors

This review on recent progresses on Gene therapy treatments in dermatological diseases presented different strategies in gene therapy and how they could be applied to various dermatological conditions. A few comments for the authors are listed below:

(1) While the introduction to gene therapy in dermatology in the manuscript is well-organized, a brief overview of traditional treatments and their shortcomings compared to gene therapy could be beneficial for a broader audience.

(2) As the authors introduced many cutting-edge gene therapy strategies that have not been FDA approved yet, like the siRNA-based treatment, more information on any preclinical or in vivo study progress would be helpful.

(3) For Table 2, if the references can be categorized based on different gene therapy treatment types among each disease, it would be easier for the readers to look into references for specific treatment.
